# The Impact of STEM Activities on Social Skills and Emotional–Behavioral Outcomes in Students with Autism Spectrum Disorder

**DOI:** 10.3390/bs15111520

**Published:** 2025-11-08

**Authors:** Fang Da, Ying Ma, Wenya Wang, Weiyu Li, Qiang Guo, Tingzhao Wang

**Affiliations:** 1Faculty of Education, Shaanxi Normal University, 199 Changannan Rd., Xi’an 710062, China; dafang@snnu.edu.cn (F.D.); maying1927@snnu.edu.cn (Y.M.); wangwenya@snnu.edu.cn (W.W.); 2Xi’an Qizhi School, 7 Daxuenan Rd., Xi’an 710068, China; liweiyu09@sina.cn; 3College of Child Development and Education, Zhejiang Normal University, 1108 Gengwen Rd., Hangzhou 311231, China

**Keywords:** STEM, project-based teaching, ASD, social skills, emotional–behavioral outcomes, single-case design research

## Abstract

Students with autism spectrum disorder (ASD) often lack the social skills required for interpersonal interactions, highlighting the urgent need for evidence-based intervention programs. STEM activities that emphasize collaboration and communication offer a new pathway for social skill development. This study developed an adaptive STEM project-based learning instructional framework teaching model and employed a multiple-probe across-participant design to evaluate the participants’ social skills achievement rates and frequency of emotional and behavioral incidents. The results indicated that STEM activities exerted positive intervention effects; they effectively improved social skills (including cooperation, empathy, engagement, and communication) in students with ASD and reduced the occurrence of emotional and behavioral problems. Feedback from teachers, parents, and students further confirmed the social validity of STEM activities. Finally, recommendations for implementing STEM education among students with ASD are proposed from three perspectives: constructing interdisciplinary collaboration mechanisms, developing adaptive STEM curricula, and implementing dynamic teaching support strategies.

## 1. Introduction

Autism spectrum disorder (ASD) is a neurodevelopmental disorder characterized by persistent deficits in social communication, restricted interests, and repetitive behavioral patterns ([3]). Over the past decade, inclusive educational practices for children with ASD have gained prominence, reflecting the growing emphasis on safeguarding their right to access mainstream classrooms ([1]). This trend not only underscores the importance of addressing their academic needs but also highlights the urgency of developing effective pedagogical strategies aligned with their unique neurocognitive profiles. Within this context, the transdisciplinary teaching model has emerged as a highly promising framework for constructing authentically inclusive learning environments. This model integrates knowledge, methods, and perspectives from multiple disciplines, enabling it to both address the complex challenges in inclusive education effectively and foster deep collaboration among educators, therapists, and families. Consequently, it provides comprehensive and tailored support for diverse learners with special needs ([58]). In inclusive educational settings, the value of transdisciplinary teaching extends beyond the systematic cultivation of learners’ academic abilities. It specifically focuses on the holistic development of key competencies, including social interaction, emotion regulation, and daily living skills. This characteristic is particularly crucial for students with ASD, ultimately empowering them to integrate more effectively into mainstream school and community environments. However, despite advances, several significant challenges remain. A key barrier is the impairment in theory of mind among individuals with ASD, which compromises their ability to interpret and appropriately respond to nonverbal cues such as vocal intonation and facial expressions, thereby complicating social interactions ([32]). Additionally, difficulties in emotion regulation frequently trigger behavioral crises, including aggressive outbursts, self-injurious behaviors, and social withdrawal, which disrupt classroom functioning and impede academic engagement ([38]; [34]). Collectively, these deficits contribute to a “social integration gap,” in which the physical inclusion of students with ASD in classrooms fails to translate into meaningful peer integration or social participation ([22]).

Building on the foundation of transdisciplinary education, science, technology, engineering, and mathematics (STEM) education provides an interdisciplinary approach that integrates inquiry-based and experiential learning, offering a unique intervention pathway for addressing social skill deficits among students with ASD ([59]). The predictable task sequences and role-defined collaborative environments inherent in STEM projects reduce open-ended social pressure. For instance, [36] ([36]) demonstrate that group-based robotics programs enhance joint attention in children with ASD, increasing cooperative behaviors, such as instrument-sharing and collaborative outcome-building, thereby circumventing ambiguities in unstructured social interactions. [41] ([41]) find that the problem-solving dependency of STEM activities naturally elicits communication attempts from students with ASD with sustained social motivation owing to direct task relevance. [9] ([9]) confirm that collaborative problem solving improves peer interaction competence and communication skills, facilitating the mastery of social rules (e.g., turn-taking and waiting) for better school adaptation. [18] ([18]) have observed that during robot assembly tasks requiring step-based component exchange, children with ASD exhibited increased directed communication (e.g., “Your turn now”) while experiencing reduced pressure for sustained eye contact.

Beyond social skills, the structured pedagogical design of STEM education is also posited to mitigate emotional–behavioral challenges in students with ASD. This structured approach not only aids in emotional regulation but has also been linked to the enhancement of core social competencies. Explicit project sequencing and visual outcome representation enhance environmental predictability, reducing uncertainty-triggered behaviors such as crying outbursts and self-stimulatory actions. This strengthened sense of environmental control diminishes anxiety responses ([52]) and creates a more stable foundation for social engagement. For instance, [63] ([63]) have documented in collaborative robotics projects that students with ASD accumulate positive interaction experiences through repeated engagement in structured group tasks. During conflict-mediation scenarios, participants practiced emotion recognition and expression, demonstrating significant improvements in their affective interpretation capacity and behavioral self-regulation. Similarly, a recent study by [46] ([46]) on engineering design challenges reported a marked increase in prosocial behaviors, such as collaborative problem-solving and peer support, among adolescents with ASD. In contrast, digitally integrated STEM tools provide innovative emotion regulation support. Technologies such as video modeling and virtual reality social scenario simulations transform abstract academic concepts into tangible visual representations, effectively reducing the cognitive load. This facilitates emotional stabilization through mastering experiences ([23]; [21]) and has been shown to improve social initiations and responses in simulated peer interactions ([30]). [45] ([45]) further substantiate that integrating natural language processing models within STEM projects provides real-time linguistic prompts and affective scaffolding. This intervention reduced anxiety and social withdrawal behaviors while promoting self-directed learning rhythm control.

Building upon this foundation, this study aims to extend prior research in several distinct ways. First, while previous studies have often focused on a single domain, such as robotics or general problem-based learning ([7]), this intervention employs a more comprehensive and integrated STEM-PBL curriculum that incorporates elements from multiple disciplines (e.g., simple engineering design and scientific inquiry) to provide a richer and more varied context for social collaboration. Second, unlike stand-alone STEM activities, our instructional framework is explicitly designed around and incorporates key evidence-based practices for ASD, such as visual supports, structured work systems, and peer-mediated instruction, ensuring the intervention is tailored to the specific learning needs of the participants. Finally, this study explores the feasibility and effectiveness of this integrated approach within the specific cultural and educational context of China, a setting that remains underexplored in existing literature, thereby contributing insights into its cross-cultural applicability.

Research indicates that STEM project-based learning (PBL) can simultaneously address the rigid thinking tendencies and executive function deficits commonly observed in students with ASD ([19]), providing an operational framework for enhancing their social skills and emotion regulation ([60]). We propose an integrated theoretical model (see Figure 1) wherein STEM-PBL activities function through three core mechanisms to produce synergistic improvements in social and emotional–behavioral domains. First, the structured, collaborative nature of STEM-PBL scaffolds social engagement through predictable routines and shared goals, thereby reducing anxiety associated with ambiguous social cues ([6]). For instance, scripted role-playing within engineering tasks allows students to practice perspective-taking and joint attention in low-stress scenarios, thereby improving social communication ([37]). Concurrently, STEM projects leverage the restricted interests common in students with ASD as motivational anchors, transforming them into collaborative opportunities and demonstrating increased spontaneous initiation and sustained social participation ([50]). Regarding emotional–behavioral outcomes, the iterative problem-solving process in STEM-PBL (e.g., hypothesizing, testing, and revising) externalizes emotional regulation through concrete steps. This scaffolding helps students reframe “errors” as discovery opportunities, mitigating frustration and perseverative behavior ([26]). Executive function gains are further amplified as students organize materials, sequence tasks, and monitor timelines and skills that are directly transferable to daily living ([64]).

Building on this foundation, this study developed a specific STEM-PBL instructional framework designed to foster social interaction among students with ASD through collaborative STEM activities. Consequently, this study aims to investigate the following questions: (1) Does participation in structured STEM activities lead to significant improvements in the social skills of students with ASD? (2) Can engagement in structured STEM activities effectively ameliorate emotional and behavioral challenges in this population? (3) To what extent are the improvements in social skills and emotional–behavioral outcomes maintained following the intervention? (4) To what extent do participants generalize the acquired skills to untrained settings or persons? (5) What are the social validity perceptions of the participants and teachers regarding the intervention?

## 2. Methods

### 2.1. Participants

Three Chinese adolescents with ASD were recruited from a special education school in Xi‘an. All participants met the following inclusion criteria: (1) Clinically diagnosed with ASD according to the Diagnostic and Statistical Manual of Mental Disorders, Fifth Edition by licensed psychologists or physicians with extensive early identification experience; (2) Scored between 35 and 69 on the Wechsler Intelligence Scale for Children—Fourth Edition (WISC-IV); (3) Absence of comorbid neurological abnormalities and no recent psychotropic medication use; (4) Normal vision and hearing, with basic verbal expression, auditory comprehension, and cognitive understanding abilities; (5) Expressed interest in pursuing STEM-related careers (mathematics, science, engineering, or technology); and (6) Provided written informed consent prior to participation. Subsequently, teachers completed the Social Skills Improvement System-Rating Scale (SSIS-RS) ([16]) to assess social skill deficits and used the Childhood Autism Rating Scale (CARS) ([40]) to evaluate autism symptom severity.

Participant 1 is a 14-year-old male eighth grader with a WISC-IV score of 60. Their CARS and SSIS-RS scores were 34 and 78, respectively. This participant demonstrates initiative in communication attempts and engages in simple social interactions (e.g., greetings/farewells) with teacher prompts. They exhibit frequent echolalic vocalizations and unprompted laughter during class.

Participant 2 is a 13-year-old male seventh grader with a WISC-IV score of 45. Their CARS and SSIS-RS scores were 38 and 70, respectively. They show minimal communicative initiation with verbal expressions largely restricted to stereotyped phrases that lack functional purposes. They use single-word requests (e.g., “Want”) when prompted, and they display recurrent self-stimulatory behaviors, including standing abruptly, repeated shoe-tying, hand flapping, and humming during instruction.

Participant 3 is a 12-year-old male sixth grader with a WISC-IV score of 54. They obtained a score of 34 on the CARS and 73 on the SSIS-RS. They avoid social interaction initiation and only notice peers after repeated teacher prompts. They demonstrate emotional dysregulation when redirected (crying outbursts, object throwing, aggression toward teachers/peers).

### 2.2. Peer Partners and Training Procedures

Each participant with ASD was paired with two same-class peers (six peer partners in total). Peer selection was based on teacher recommendations, with primary consideration given to the following practical behavioral indicators: (1) the ability to engage in basic social interactions (e.g., turn-taking, sharing materials); (2) relatively stable emotional and behavioral regulation, with no significant aggressive or severely disruptive behaviors; (3) a school attendance rate greater than 95%; and (4) provision of informed consent by their parents and assent by the peers themselves.

One week prior to the intervention, we conducted three structured preparation sessions for the peers, each lasting 40 min. These sessions focused on behavioral rehearsal and visual support, rather than conceptual psychoeducation, and were led by the principal researcher with assistance from the classroom teacher.

Session 1: Learning Our “Jobs”. This session introduced the peer role through concrete actions and visual scripts. Using an “I do, We do, You do” modeling sequence, peers were taught specific, observable behaviors. Key skills included how to hand materials to a partner, how to indicate the next step on the task sheet, and how to use simple phrases like “Your turn.” Role-playing using actual STEM materials served as the primary teaching method.

Session 2: Practicing Helping. This session focused on developing a limited repertoire of responses to common scenarios. Peers practiced how to react when a partner was not participating (prompt: “Point to the group collaboration worksheet and say, ‘Your turn’”) and how to provide help by demonstrating rather than explaining (prompt: “If he looks confused, point to the material he needs”). Laminated visual prompt cards with icons and simple words were introduced and practiced.

Session 3: Integrated Rehearsal. The final session was a complete run-through of a simplified STEM activity. Peers used the group collaboration worksheets and visual prompt cards to interact with researchers who role-played the roles of participants with ASD. The goal was to build their fluency and familiarity with the activity flow and the support tools.

Throughout all STEM-PBL sessions, peers continuously received embedded support designed to assist them in successfully fulfilling their partner roles. This included:(1)A 3 min pre-session preview of the day’s activity using the group collaboration worksheet before each session.(2)Continuous access to visual prompt cards printed with simple cooperation icons and keywords (e.g., “Take turns,” “Help,” “Share,” “Good job”), which both peers and participants could refer to during interactions.(3)Discreet, direct behavioral prompting from researchers and teachers (e.g., if a participant needed help but the peer did not notice, the researcher would quietly prompt the peer: “He seems to be having difficulty, you could ask him ‘Do you need help?’”).(4)Specific, behavior-contingent praise for peers (e.g., “You did a great job just now taking the initiative to share the tool!”), to reinforce the desired supportive behaviors.

### 2.3. Setting

All collaborative group sessions were conducted in the laboratory of the Autism Research Center at a special education school in Xi’an, where the participants with ASD and their peer partners jointly completed STEM activities. The laboratory is spacious, quiet, and noise-free, with a maximum capacity of 20 people. It was equipped with three square tables with drawers, each paired with four chairs. Additionally, the network facilities were well equipped to ensure stable and smooth network connectivity during the experiment. The drawers of the tables contain group cooperation task sheets as action guides, as well as most of the required materials (such as glue, tape, scissors, pencils, watercolor pens, erasers, strings, and cotton).

### 2.4. Materials

The STEM activities utilized the Banma^©^ Science L1 STEM Exploration Kit as the primary instructional material. Collaborative task sheets (Table 1) guided the student groups through a step-by-step inquiry process. Content delivery and assessment were conducted using Huawei MatePad 11 tablets that displayed thematic picture books, experimental demonstration videos, and evaluation tools. The intervention sessions were recorded using a digital video camera (Sony FDR-AX43, Sony Corporation, Xi’an, China) to ensure accurate behavioral coding. All instructors were certified special education teachers with ≥3 years of experience teaching students with ASD.

### 2.5. Independent Variable

The independent variable in this study was the adapted STEM-PBL instructional framework designed for students with ASD. This model integrates the four-phase STEM-PBL instructional framework proposed by [12] ([12]) and structured as follows: (1) goal anchoring, (2) collaborative task design, (3) guided engineering inquiry, and (4) structured reflection (see Table 1).

#### 2.5.1. Explicit Instruction

Utilizing ABA-informed prompting techniques within a multi-scaffolding approach, instruction was delivered through guided student practice and teacher feedback ([15]). This methodology has been established as an evidence-based practice for science inquiry among students with ASD ([54]). The Model-Lead-Test paradigm has been successfully implemented for teaching mathematics and science skills to ASD learners ([27]). Such a modified constructivist pedagogy enables students with ASD and other developmental disabilities to achieve success in STEM practices ([29]). This study integrated direct instruction with complementary pedagogical strategies to effectively guide ASD students’ engagement in collaborative groupwork.

#### 2.5.2. Task Analysis Lists

Teachers provided students with step-by-step checklists outlining the task procedures, enabling the sequenced completion of learning activities. This approach significantly reduces the working memory load for students with cognitive disabilities and effectively scaffolds scientific inquiries ([14]). For instance, [42] ([42]) have implemented self-monitoring checklists during scientific investigations to guide problem-solving. Students verbally articulated the completed steps using checklist prompts while conducting self-assessments. Building on self-monitoring frameworks, this study developed a collaborative group task sheet that simplified projects into three to four structured subtasks, incorporated role-based collaboration protocols and phase-driven evaluation rubrics, and systematically guided students through STEM activities. Figure 1 illustrates a sample task sheet (see Figure 2).

#### 2.5.3. Video Prompting

Target skills or behaviors were segmented into step-by-step video clips, allowing learners to practice each step immediately after viewing, before progressing sequentially ([62]; [5]). Video prompting imposes relatively low demands on attention and working memory, aligning with the cognitive and visual learning strengths of students with ASD ([56]). For example, [61] ([61]) demonstrate the efficacy of video prompting in teaching programming skills to students with ASD, noting enhanced skill generalization. Video prompting also improves group learning outcomes as evidenced by [43]’s ([43]) study of chained task instructions. Consequently, this study employed video prompting to provide visual scaffolding during STEM instruction for students with ASD.

### 2.6. Dependent Variable and Measurement

The dependent variables were the participants’ social skills and emotional–behavioral outcomes, which are operationally defined in Table 2. The SSIS-RS Teacher Form was administered. Four target dimensions were confirmed through the triangulation of parent-teacher interviews and observational data: cooperation, empathy, participation, and communication ability (26 items in total). The scoring protocol was as follows: 0 = absent (not observed); 1 = prompt-dependent (requires continuous physical or verbal guidance); 2 = cue-dependent (requires intermittent verbal prompts); and 3 = self-initiated (spontaneously exhibited). Total scores ranged from 0 to 78. The skill attainment rate was calculated using the formula: “Target skill achievement rate = (Actual score/78) × 100%.” Emotional and behavioral problems in individuals with ASD are essentially failures in emotional regulation, manifested as maladaptive strategies that can be systematically categorized into two major types: internalizing and externalizing problems ([39]). Internalizing problems refer to internally directed maladaptive expressions, including social withdrawal, anxiety/depression, and stereotypical behaviors associated with executive function deficits and difficulties in emotion recognition in individuals with ASD. Externalizing problems manifest as overt behavioral dyscontrol, such as aggressive behaviors, attention deficits, and emotional outbursts, which are primarily related to impulse control disorders and abnormal sensory processing ([4]). Target behaviors were recorded using partial interval recording during 1 min observation windows spaced 10 min apart. A behavior was scored as occurring if it was observed at any point during the 1 min interval.

### 2.7. Experimental Design

This study employed a multiple-baseline across-participants single-case experimental design ([17]) to evaluate the effects of the STEM-PBL intervention. The design involved the sequential introduction of the intervention across three participants after establishing a stable baseline for each. Data were collected continuously across all phases (baseline, intervention, and maintenance) for all participants. Following established single-case research standards ([31]), the intervention was introduced to the first participant only after a stable baseline pattern was established (i.e., five consecutive stable data points). The intervention was then introduced to the second participant after two conditions were met: (1) the first participant demonstrated a clear intervention effect, and (2) the second participant’s baseline data remained stable (i.e., at least three consecutive data points showing a non-improvement trend). This same staggered introduction procedure was replicated for the third participant. All participants entered the maintenance phase after achieving the performance criterion (≥80% target achievement) across three consecutive intervention sessions. Visual and statistical analyses of the continuous data across phases were conducted to examine the functional relation between the intervention and the outcomes.

### 2.8. Procedures

This study was led by a doctoral candidate in special education with nine years of ASD teaching experience and a STEM background, who conducted core intervention implementation, data collection, and participant training. During the intervention sessions, two master-level special education assistants were positioned 1.5 m behind the primary intervener to minimize interference. Prior to study commencement, both assistants underwent standardized protocol training: Assistant 1 received instructions in prompt hierarchies and reinforcement procedures—utilizing verbal/nonverbal prompts (e.g., gestures, facial expressions) and reinforcements (e.g., verbal praise, nodding, high-fives)—to foster collaborative behaviors, including proactive conversation initiation, clarification-seeking, and peer support provision; both assistants were trained to master criteria (100% procedural fidelity across two consecutive role-play scenarios). Assistant 2 served as an independent observer, documenting emotional–behavioral incidents and scoring target social skills at each phase conclusion to ensure treatment fidelity.

#### 2.8.1. Baseline

During baseline sessions, the researcher presented STEM materials (e.g., scientific inquiry kits, task sheets, tablets) in an environment identical to the intervention setting, randomly assigning starting group members before initiating a 30 min naturalistic observation period; if participants engaged in solitary task operation, only a single non-reinforcing prompt (e.g., “Please attempt collaboration”) was provided, with no additional instruction permitted. Baseline probes were conducted on alternate days using a multiple-probe design with staggered randomization to collect ≥5 data points per participant, continuing until three consecutive data points demonstrated stability or a downward trend with no upward pattern, at which point the participant transitioned to the intervention phase. All sessions were video recorded using dual cameras for subsequent behavioral coding and fidelity analysis.

#### 2.8.2. Intervention

The researcher implemented the STEM-PBL instructional framework for the three participants at a frequency of three 40 min sessions per week, structured into four evidence-based stages following standardized protocols: (1) Goal anchoring: Establishing core project tasks and social interaction objectives through group discussion to orient collaboration; (2) Collaborative task design: Introducing an innovative role-rotation system where participants cycled through three structured roles—observer (documenting experimental data/peer interactions), operator (executing core tasks/initiating help-seeking), and material manager (distributing resources/proactively offering support)—following an “observation → imitation → practice” progression to scaffold social skill development from observational learning to proactive assistance; (3) Guided engineering inquiry: Utilizing video prompting and demonstration videos; after viewing 30 s exemplars, participants performed tasks independently, with peer-assisted completion triggered after three consecutive errors, while the intervener employed standardized verbal prompts (e.g., requiring operators to use scripted requests: “I need [material], please assist”); (4) Structured reflection: Facilitating systematic debriefing via guided questioning (“Specific role responsibilities,” “Problems encountered/solutions,” “Product functionality,” “Knowledge/skill gains”) to enhance metacognitive processing of collaboration. Multidimensional data collection included target skill rating scales, event sampling of emotional–behavioral incidents, tablet-based interdisciplinary assessments, and satisfaction surveys with immediate positive reinforcement of prosocial behaviors. Stage advancement required achieving ≥80% target skill accuracy across three consecutive sessions (see Figure 3).

#### 2.8.3. Generalization

Following the intervention phase, all procedures were withdrawn during the maintenance phase with assessment methods identical to the baseline conditions. Target skill retention was evaluated through six data collection points commencing one week post-intervention. The measurement instruments included (a) the Target Skills Rating Scale adapted from the SSIS-RS to record attainment percentages across cooperation, empathy, engagement, and communication domains; (b) the Emotional–Behavioral Incident Log to document frequency counts of emotional–behavioral occurrences (each incident scored as +1); and (c) the Intervention Procedure Checklist auditing implementation accuracy across critical dimensions (intervention setting preparation, material availability, and adherence to verbal prompting protocols). All instruments demonstrated adequate content validity through expert reviews and pilot testing.

### 2.9. Interobserver Agreement and Procedural Reliability

To ensure data objectivity and validity, the third and fourth authors independently scored 40% of randomly selected video recordings from each phase (baseline, intervention, and maintenance) using dependent variable measurement protocols. Interobserver agreement (IOA) was calculated using the interval-by-interval method ([17]), in which the number of agreed intervals was divided by the total number of intervals (agreement and disagreement) and multiplied by 100%. For social skills outcomes, the third and fourth authors independently scored 40% of randomly selected video recordings from each phase, achieving 100% IOA across all participants and phases using the interval-by-interval method. For emotional–behavioral outcomes, a trained doctoral student who was blind to the study phases independently coded 30% of sessions; comparison with the primary observer’s (Assistant 2) records yielded a mean IOA of 96.3%. Procedural reliability was assessed using the Intervention Procedure Checklist developed for this study; the second author reviewed 30% of randomly sampled sessions per participant, calculating fidelity as: (number of steps implemented as planned ÷ total steps) × 100%. The resulting fidelity coefficients were 98.8%, 100%, and 95.3% across participants, respectively, confirming strict adherence to the prescribed intervention protocol.

### 2.10. Social Validity

To comprehensively assess the social validity of the STEM-PBL intervention, semi-structured interviews were conducted with multiple stakeholders, including the three student participants, their homeroom teachers (n = 3), and their parents (n = 3). During the pre-intervention phase, interviews with teachers and parents were held to establish socially significant target skills by gathering detailed reports on the participants’ school-based social interactions and emotional expression patterns. Following the intervention, all stakeholder groups were interviewed again to evaluate the intervention’s acceptability and perceived outcomes: teachers and parents assessed its appropriateness, feasibility, and perceived changes in student skills, while student participants reported on their satisfaction with the materials, activities, and overall experience. To analyze the interview transcripts, we employed thematic analysis following the systematic process outlined by [10] ([10]), which involved data familiarization, code generation, and theme development and review. Furthermore, investigator triangulation was utilized to enhance credibility, whereby two researchers independently coded a portion of the transcripts and then discussed discrepancies to reach a consensus.

### 2.11. Data Analysis

Visual analysis is the foundation of data analysis in this single-case research design to determine whether a causal relationship exists between independent and dependent variables ([33]). We adopted a systematic visual analysis process to identify causal associations between the intervention measures and the target skills. The study employed visual analysis to examine the trends and stability of participants’ target social skills, as well as emotional and behavioral outcomes. Complementing the visual analysis, the Tau-U effect size ([47]) was calculated to quantify the intervention effects across all three participants, integrating non-overlap with baseline trend control to provide a robust effect magnitude metric.

## 3. Results

### 3.1. Social Skills Intervention Outcomes

A consistent pattern of improvement in social skills was observed across all three participants following the implementation of the STEM-PBL instructional framework. Visual analysis of the graphed data (Figure 4) and statistical analyses revealed substantial increases in target skill attainment from low and stable baselines to the intervention phases, with skills stabilizing after 18–23 sessions. Tau-U effect sizes indicated strong to perfect intervention effects, with maintenance phases demonstrating strong persistence of acquired skills. Detailed phase-level data for each participant are summarized in Table 3, Table 4 and Table 5.

Participant 1. Visual analysis revealed a clear functional relationship between the intervention onset and skill acquisition. Baseline performance was stable with no mastery, which sharply contrasted with the immediate and substantial improvement upon intervention introduction. These results demonstrate the successful generalization and maintenance of Participant 1’s four target social skills (see Table 3).

Participant 2. As illustrated in Figure 4, Participant 2’s data exhibited a similar, though initially more variable, ascending trend during the intervention (Figure 4). The performance criterion (≥80%) was met and sustained, with maintenance phase performance further improving, confirming robust and sustained skill generalization (see Table 4).

Participant 3. As depicted in Figure 4, Participant 3 also demonstrated a marked improvement following the intervention’s onset, reaching the performance criterion and maintaining gains. The data path supports the effectiveness of the intervention for this participant across all target social skills (see Table 5).

Tau-U effect sizes confirmed these visual analyses. Participants 1 and 3 showed perfect effects across all skills (Tau-U = 1.00, 95% CI [0.63, 1.00]). Participant 2 showed perfect effects for engagement and communication (Tau-U = 1.00, 95% CI [1.00, 1.00]), a near-perfect effect for cooperation (Tau-U = 0.95, 95% CI [0.71, 1.00]), and a large effect for empathy (Tau-U = 0.83, 95% CI [0.64, 1.00]). According to [49] ([49]), Tau-U values above 0.92 represent strong effects, indicating a highly effective intervention. Maintenance effects were perfect for all participants and skills (Tau-U = 1.00, 95% CI [0.29, 1.00]).

### 3.2. Emotional–Behavioral Intervention Outcomes

Visual analysis indicated that the STEM-PBL framework led to immediate and sustained reductions in emotional–behavioral problems across all three participants (see Figure 5). Following a stable baseline with high frequencies of incidents, the intervention phase showed a clear and steady downward trend, which stabilized further at low levels during maintenance.

The robust visual analysis findings were statistically corroborated by Tau-U analyses. A consistent and strong decreasing trend in target behavior from the baseline to the intervention phase was confirmed across all participants, as evidenced by a baseline-corrected Tau-U value of −1.00 (95% CI [−1.00, −0.33]). In line with the benchmarks established by [49] ([49]), an absolute Tau-U value of 1.00 indicates 100% non-overlap between the baseline and intervention phases, reflecting complete separation of phase-specific data points and representing the strongest possible magnitude of intervention effect. The negative sign of the Tau-U value further confirms that this robust effect aligned with the intended direction of change—specifically, a reduction in the frequency of the target incidents. Furthermore, when comparing the maintenance phase to the baseline, the effect size remained at −1.00. This result demonstrates that the substantial, therapeutically meaningful reduction in target behavior achieved during the intervention phase was fully sustained over the maintenance period, with no evidence of behavioral regression.

These results indicate that the STEM-PBL teaching framework had a relatively significant control effect on the emotional and behavioral outcomes of all three participants, as well as a positive impact on reducing problematic emotional and behavioral incidents.

### 3.3. Social Validity

The thematic analysis of post-intervention interviews with teachers, parents, and students revealed three central themes regarding the social validity of the STEM-PBL intervention: (1) perceived improvements in participants’ social-communication skills, (2) observed growth in emotional and behavioral regulation, and (3) overall stakeholder endorsement of the intervention’s feasibility and appeal.

#### 3.3.1. Theme 1: Enhancement of Social-Communication Skills

All teachers and parents reported observable gains in the participants’ social initiation and communication. For instance, Participant 1’s teacher noted a marked increase in peer awareness and proactive helping behaviors, such as initiating help with phrases like, “Nana, do you need my help?” Similarly, the parents of Participant 2 observed the child beginning to initiate social interactions outside of school, such as sending WeChat invitations to friends for offline activities.

#### 3.3.2. Theme 2: Improvement in Emotional and Behavioral Regulation

Stakeholders highlighted significant progress in the participants’ ability to self-regulate. The homeroom teacher for Participant 2 emphasized a “significant improvement in rule awareness,” including increased patience in waiting for materials. For Participant 3, parents reported an increased duration of joint attention and an improved ability to sit and complete tasks for extended periods, indicating better emotional and behavioral control.

#### 3.3.3. Theme 3: High Feasibility and Acceptability of the Intervention

The STEM-PBL framework was unanimously deemed feasible and effective by the educators. Furthermore, the intervention was highly engaging from the students’ perspective. All three participants expressed that the STEM activities were highly engaging, highlighting the relevance of projects and materials to their daily lives. They asked intervention implementers about the possibility of continuing activities post-intervention, demonstrating strong approval.

## 4. Discussion

This study, based on three participants with ASD, implemented an integrated STEM-PBL instructional framework that embedded multiple evidence-based interventions, demonstrating robust implementation outcomes. The findings from these cases provide strong initial evidence for the efficacy of this approach. Significant improvements were observed across four core skill domains (cooperation, empathy, engagement, and communication) in all three participants with ASD. These findings align with [61]’s ([61]) research, which confirms that multicomponent frameworks incorporating visual support and direct instruction effectively promote STEM engagement and social behavior development in students with ASD.

Within the current framework, structured task sheets enable participants to conduct collaborative inquiries through systematic task analysis procedures, facilitating the comprehension of group roles, turn-taking protocols, and operational sequences, and enhancing academic concept mastery and social skill acquisition in ASD learners ([28]). Further supporting the framework’s efficacy, [55] ([55]) establishes that explicit procedural scaffolding, a core element of the STEM-PBL instructional framework, enables the implementation of PBL among children with intellectual disabilities. The embedded video prompting component specifically addresses the attentional and executive function challenges inherent to ASD by reducing the cognitive load while increasing instructional efficiency ([25]). Within the framework’s peer-mediated cooperative structure, participants assimilated the observed social behaviors through cognitive-behavioral integration. Teacher-mediated prompts and reinforcement facilitated the progressive internalization of these skills, which is consistent with [8]’s ([8]) evidence that observational learning within structured frameworks advances functional communication among neurodiverse learners.

The implementation of this comprehensive framework within the context of Chinese education presents both advantages of cultural adaptability and challenges stemming from practical conditions. The emphasis on collective collaboration and the dual-subject (teacher-student) dynamic in Chinese culture aligns with the collaborative needs of STEM-PBL, which emphasizes peer mutual assistance and teachers’ scaffolding support. However, resource constraints in special schools and large class sizes in mainstream settings may affect implementation fidelity. These cultural and contextual factors should be considered when evaluating the generalizability of our findings to other educational systems.

Beyond investigating social skills, this study examined emotional–behavioral outcomes in students with ASD. The experimental results indicated a reduction in the frequency of emotional–behavioral problems across all three participants, with stable maintenance effects. According to the Predictive Impairment in Autism hypothesis, individuals with ASD exhibit a diminished capacity for predictive coding of event contingencies ([53]). Environmental unpredictability may consequently provoke anxiety, potentially contributing to emotional–behavioral dysregulation. Crucially, ritualized procedural invariants can mitigate anxiety by enhancing predictive stability ([57]). Within our STEM-PBL instructional framework, the implementation of structured collaborative task sheets provided anticipatory scaffolding for sequential actions. This procedural predictability likely enhances psychological safety, constituting one mechanism for reducing emotional–behavioral incidents. Furthermore, embedded behavioral strategies, including video prompting and direct instruction, have progressively introduced novel sensory inputs through systematic repetition. This aligns with Bayesian accounts of ASD, suggesting that such techniques reduce environmental uncertainty by reinforcing predictive models ([20]), thereby decreasing emotional–behavioral frequency. Empirical support exists for video-based interventions that specifically ameliorate emotional dysregulation in individuals with ASD ([2]). STEM activities cultivate an affectively supportive learning ecology ([44]), transforming participants from passive recipients to active agents. Engagement fosters self-efficacy through mastering experiences. Notably, researchers have observed increased joint attention duration and enhanced task persistence during STEM activities, which are factors that potentially contribute to reduced emotional–behavioral occurrences through improved cognitive engagement and self-regulation capacity.

This study extends the scope of STEM education research to students with ASD, with particular focus on those presenting with co-occurring moderate intellectual disabilities. By adapting core PBL principles, we developed a specialized STEM-PBL instructional framework tailored to learners with ASD. This approach established a viable implementation pathway for social skills interventions in this population. The interdisciplinary nature of STEM education enables exceptional learners to construct an authentic, contextualized understanding of the world, transcending the limitations of isolated fact-based instruction ([11]). This pedagogical imperative underscores the need to prioritize STEM accessibility for neurodiverse learners with specialized educational requirements. Consistent with inclusive education research, [35] ([35]) emphasize that effective STEM instruction necessitates proactive pedagogical adjustments based on a comprehensive understanding of students’ learning barriers. Furthermore, [48] ([48]) address collaborative learning disadvantages by designing self-advocacy-focused STEM approaches that significantly enhanced academic outcomes for students with attention deficits and specific learning disabilities.

To scale this intervention in mainstream classrooms, we recommend (1) developing a tiered implementation model with differentiated tasks (e.g., basic, advanced, extended) to accommodate students with varying levels of need; (2) utilizing intelligent learning systems to provide personalized visual supports (e.g., step-by-step illustrated task cards, voice-prompted guides) and leveraging assistive technologies to reduce operational barriers; (3) establishing a co-teaching framework where general education teachers lead instruction while special education teachers provide embedded support, and organizing heterogeneous small groups with trained peer buddies to facilitate social integration. These findings underscore the need to incorporate training on evidence-based practices for ASD into STEM teacher development programs and to advocate for policies that support the creation of adaptive STEM resources.

### 4.1. Limitations and Directions for Future Research

This study had three primary limitations. First, the participant cohort focused exclusively on students with ASD and co-occurring moderate intellectual disabilities. Given that varying combinations of intellectual disability severity and ASD symptomatology may differentially impact intervention outcomes, the findings cannot be generalized to those with mild intellectual disability or ASD without comorbidities, nor can they address the needs of neurotypical students. Moreover, the absence of long-term efficacy tracking (>6 months) limits its broader applicability. Furthermore, the findings must be interpreted within the context of the intensive multiple-baseline single-case design. While this approach provided robust experimental control and rich, case-specific insights into the effects of the intervention for the three students with ASD, the small number of participants inherently limits the immediate generalizability of the results. Future research with larger sample sizes is necessary to establish the broader efficacy of this STEM-PBL framework.

Second, reliance on teacher-completed rating scales for social skills assessment introduces potential rater bias and insufficiently validates skill generalization. This singular data source may yield subjective interpretations that fail to capture students’ authentic competency levels comprehensively.

Most critically, curricular resource limitations may compromise intervention depth and breadth. Although adaptations of the Banma^©^ Science L1 STEM program enhanced operational feasibility through task decomposition and stepwise scaffolding, significant deficits persist in cross-disciplinary integration and socio-contextual diversity.

Based on these findings, we propose three systematic recommendations, the top priority of which is establishing an interdisciplinary collaboration mechanism. Schools should integrate ASD rehabilitation specialists, behavioral analysts, and STEM teachers to form teaching and research teams. The teams should focus on establishing student development files and a baseline database of STEM abilities and developing a stepped PBL guidebook. This guidebook should clarify visual prompting strategies, environmental adaptation plans, and peer collaboration support frameworks for different ability levels at each stage of the project.

Second, adaptive STEM curricula should be developed. Priority should be given to designing engineering projects in social contexts (such as the improvement of community-accessible facilities) to stimulate ASD students’ awareness of social consciousness by solving real-world problems. During task decomposition, visual processing strategies should be adopted to transform STEM project steps into visual cognitive scaffolds such as flowchart illustrations and 3D modeling.

Finally, in the process of collaborative inquiry, in addition to structuring the interaction process through tools such as role assignment worksheets and flowcharts, teachers should simultaneously incorporate communication skills training to cultivate students’ abilities in two-way dialogue and emotional regulation. It is worth noting that future STEM teaching could also introduce multimodal emotion recognition technology to monitor students’ anxiety levels and adjust task difficulty in a timely manner.

### 4.2. Conclusions

This study investigated the impact of STEM activities on social skills and emotional–behavioral outcomes in students with ASD using an embedded multi-strategy STEM-PBL instructional framework. The findings demonstrate that the targeted STEM-PBL intervention had significant positive effects on social skill development, with robust improvements observed across four core domains: cooperation, empathy, engagement, and communication. Concurrently, the intervention exerted positive moderating effects on the reduction in emotional–behavioral problems.

Multi-informant perspectives (teachers, parents, and participants) confirmed successful skill generalization in daily contexts, establishing compelling social validity for this approach. The findings substantiate that integrating structured support within STEM-PBL creates meaningful skill acquisition scenarios in which authentic problem solving facilitates both skill internalization and practical application, thereby providing an empirically grounded and practically applicable intervention model for socio-emotional development in ASD populations.

The in-depth investigation of these three cases through a rigorous multiple-baseline single-case experimental design provides strong initial evidence for the efficacy of the STEM-PBL approach. Future research should build upon these findings by implementing large-scale group studies to consolidate the evidence and explore its broader applicability.

## Figures and Tables

**Figure 1 behavsci-15-01520-f001:**
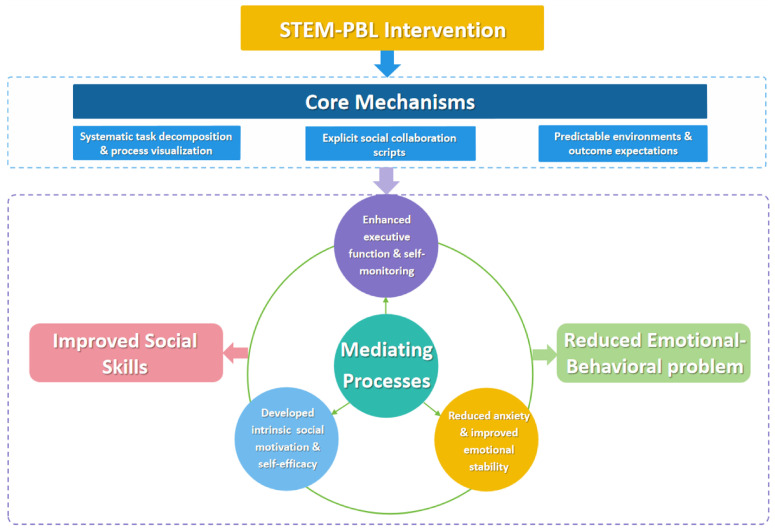
Theoretical Model of STEM-PBL Intervention for ASD Students.

**Figure 2 behavsci-15-01520-f002:**
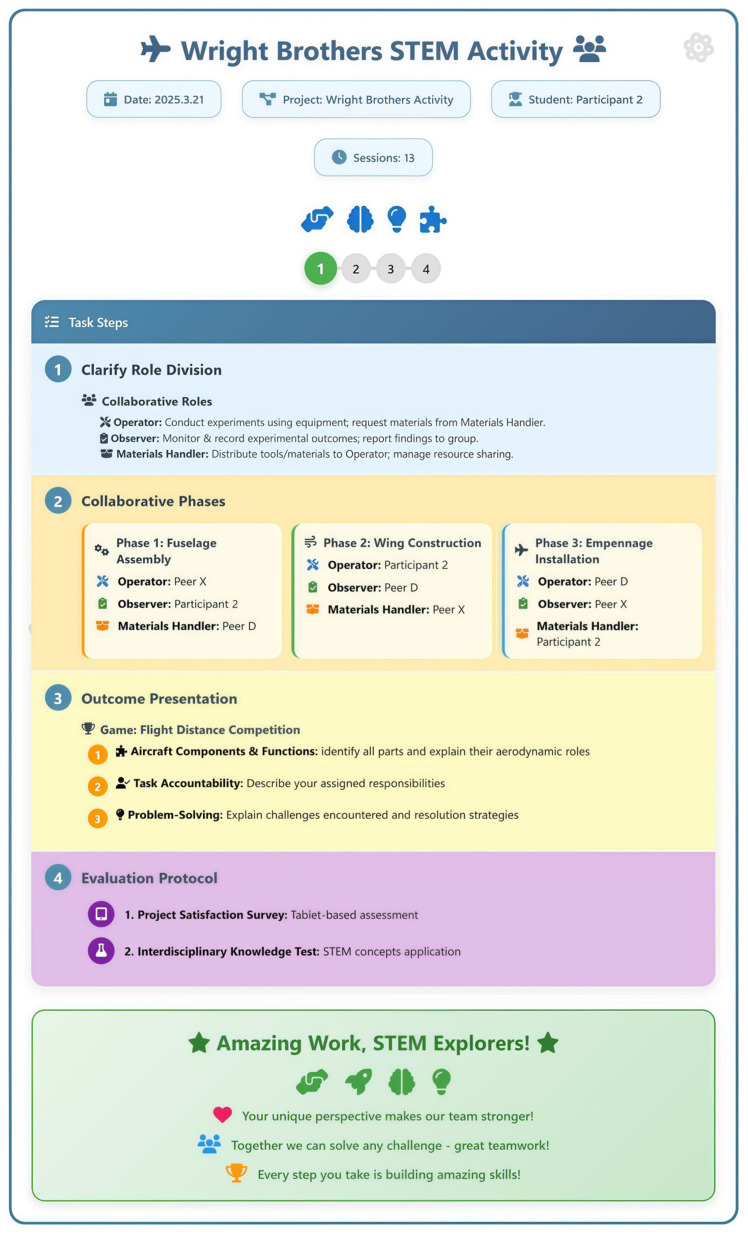
Sample of collaborative group task sheet.

**Figure 3 behavsci-15-01520-f003:**
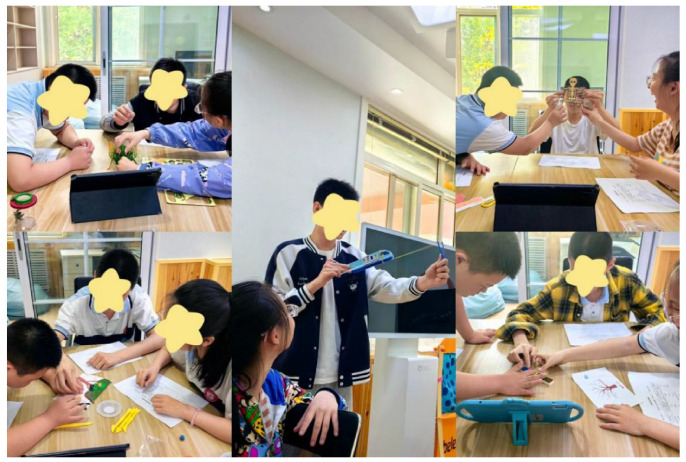
Sample of STEM-PBL group collaboration.

**Figure 4 behavsci-15-01520-f004:**
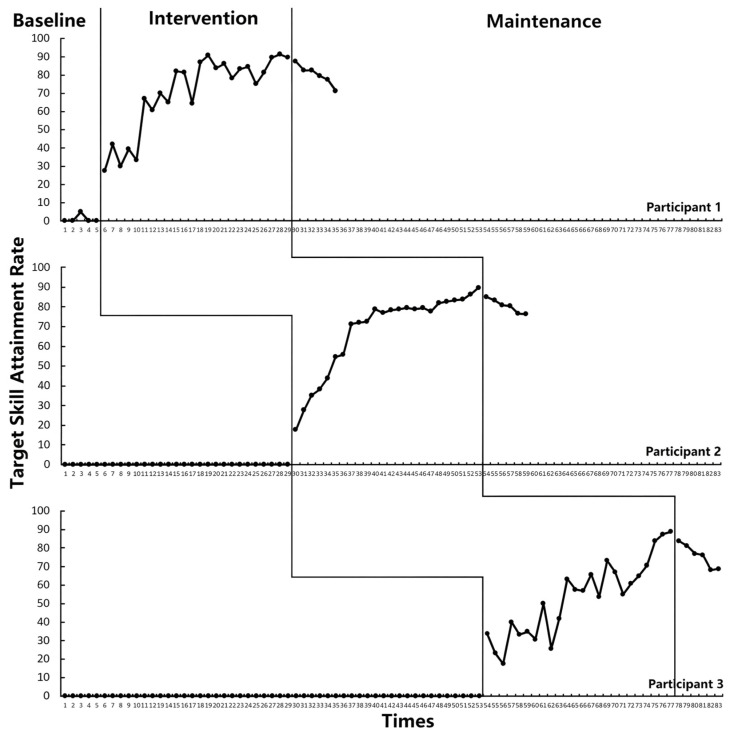
Variation trend of target skills in the three participants.

**Figure 5 behavsci-15-01520-f005:**
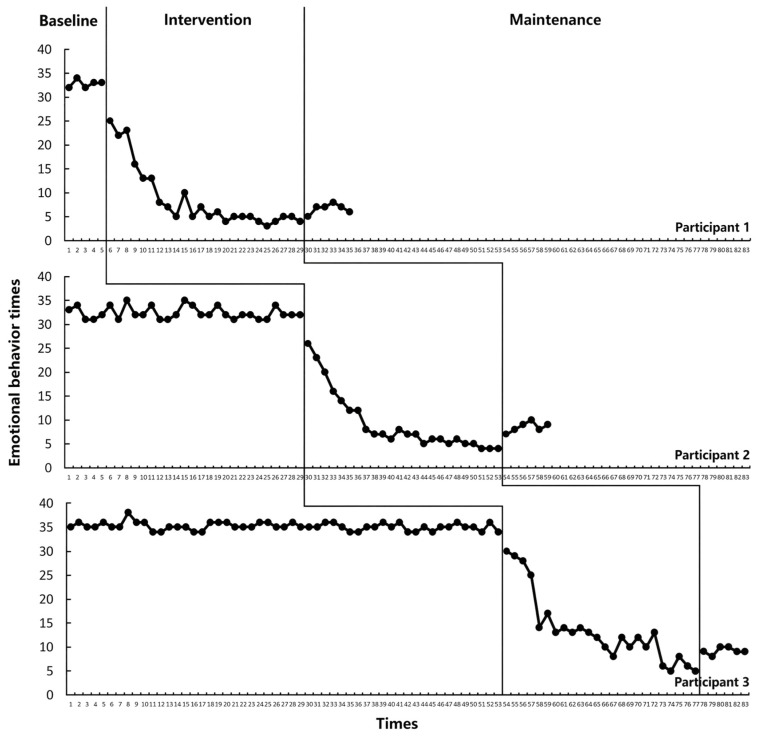
Variation in emotional–behavioral outcomes across phases for the three participants.

**Table 1 behavsci-15-01520-t001:** STEM-PBL instructional framework for students with ASD.

Phase	Core Procedure	Theoretical Underpinning
Goal Anchoring	Project selection emphasized authentic real-world problems. Teachers facilitated group discussions to identify problems, thereby compensating for ASD students’ impairments in cognitive flexibility during information integration ([13]).	Executive Function Theory
Collaborative Task Design	The project was simplified into three to four subtasks with accompanying design and implementation scripts ([24]). Structured worksheets provided explicit guidance to reduce cognitive load associated with attention, memory, sequencing, and organization in the ASD students ([51]).	Cognitive Load Theory
Guided Engineering Inquiry	Dual Scaffolding Support System: (1) Direct Instruction in Social Strategies: Teachers explicitly articulated role allocation mechanisms within worksheets and delivered targeted instruction on essential social strategies to ASD students. (2) Digital Assistive Tools: Structured worksheets, video prompts, and AI conversational agents scaffolded collaborative problem-solving processes.	Scaffolding Theory
Structured Reflection	Multi-Dimensional Assessment Framework: (1) Product/Artifact Evaluation: Students were guided by teachers to demonstrate final outcomes, verbally articulating functional features of the product, collaborative challenges encountered, conflict resolution strategies employed, and individual task responsibilities. (2) Target Skill Evaluation: Teachers systematically rated and documented target social skills and emotional behaviors based on direct observation of STEM activity performance using standardized rubrics. (3) Interdisciplinary Knowledge Assessment: Students completed cross-disciplinary knowledge tests and project satisfaction surveys via tablet-based interfaces.	Metacognition Theory

**Table 2 behavsci-15-01520-t002:** Dependent variables measures.

Dependent Variables	Target Behavior	Operational Definition	Example
Social Skills	Cooperation	The ability of an individual to respond to rules and advance task progress through observable behaviors, such as complying with group instructions or sharing resources during group activities.	After a peer reminded her, “We need to take turns with the puzzle,” Helen handed her puzzle piece to the peer within 10 s, without any verbal or physical protest. This behavior was recorded as one occurrence.
	Empathy	The ability to recognize and understand others’ emotions and provide support through observable verbal statements or physical gestures.	When her peers succeeded in their tasks, Helen expressed admiration and said, “Good job”; when her peers failed, she either patted them on the back or said, “It’s okay, try again.” This behavior was recorded as one occurrence.
	Engagement	The ability to integrate into group activities by performing role-specific actions and maintaining focus on the task.	As the designated materials handler, Helen remained in her seat for the entire observation interval, referring to the task list to pick up all required materials for the activity. This behavior was recorded as one occurrence.
	Communication	The ability to clearly express intentions and respond to others through clear verbalizations, gestures, or physical assistance.	When a peer asked whether the installation direction of the toy airplane’s wing was correct, Helen could verbally express to the peer, “It should be placed here,” or help them install it correctly. This behavior was recorded as one occurrence.
Emotional- Behavioral Outcomes	Stereotyped Behaviors	Repetitive body movements serve a self-regulatory function, not the task goal.	Helen relied on shaking her body or clapping her hands to calm her emotions during activities. Repetitive clapping or body shaking occurring more than 5 times per minute was recorded as one instance.
	Social Withdrawal	Excessive worry and social avoidance, accompanied by emotion expression deficits secondary to alexithymia. The behavior of avoiding social interactions by physically orienting away from the initiator.	Following an initiation of interaction by a peer or teacher (e.g., question, invitation), Helen exhibited an active avoidance response within 10 s, including (a) turning the head more than 45 degrees away from the initiator’s face, or (b) turning the torso more than 90 degrees away. This behavior was recorded as one occurrence.
	Depression/ Anxiety	The expression of worry or desire to escape from the unpredictable social and task environment.	Following an unexpected change in the task rules, Helen asked the teacher, “Are you sure?” or “What do we do now?” more than 3 times within the 1 min observation interval. Each query was recorded as one occurrence.
	Attention Deficits	Narrow attentional breadth, poor stability, insufficient distribution ability, or difficulty in shifting attention.	During a group instruction period, if Helen turned her head away from the speaker and looked at the window for a continuous period of more than 10 s within the 1 min observation interval, it was recorded as one occurrence.
	Emotional Outbursts	When needs are frustrated or rules conflict, individuals suddenly exhibit intense, uncontrolled emotional outbursts.	When Helen’s behavior was stopped by teachers or peers during group activities, if she continued crying or throwing objects for more than 30 s, it was recorded as one instance.
	Aggression	Aggressive behaviors (self-directed or other-directed) precipitated by frustration, anxiety, or communication breakdowns.	When Helen suddenly snatched a peer’s parts and pushed them over after experiencing failure in robot assembly, causing the activity to be suspended, it was recorded as one instance.

**Table 3 behavsci-15-01520-t003:** Visual analysis of Participant 1’s social skills.

**In-Phase Analysis:**												
**Social Skill**	**Cooperation**			**Empathy**			**Engagement**			**Communication**		
Sequence	A	B	C	A	B	C	A	B	C	A	B	C
Length	5	24	6	5	24	6	5	24	6	5	24	6
Trend	-	/	\	-	/	\	-	/	\	-	/	\
Trend Stability	100.00%	45.83%	50.00%	100.00%	54.17%	50.00%	80.00%	41.67%	33.33%	100.00%	54.17%	50.00%
Average	0.00%	64.17%	76.67%	0.00%	60.63%	74.17%	2.00%	79.58%	90.83%	2.00%	76.13%	78.83%
Level Range	0–0%	10–95%	65–85%	0–0%	10–90%	60–80%	0–10%	35–100%	85–100%	0–10%	30–100%	75–85%
Level Stability	100.00%	0.00%	83.33%	100.00%	25.00%	83.33%	100.00%	29.17%	100.00%	100.00%	50.00%	100.00%
	S	US	S	S	US	S	S	US	S	S	US	S
Level Change	0–0	10–85	85–65	0–0	20–85	80–60	10–0	35–100	100–85	10–0	45–88	85–75
	(=)	(+75)	(−20)	(=)	(+65)	(−20)	(−10)	(+65)	(−15)	(−10)	(+43)	(−10)
C Value	-	0.76	0.68	-	0.85	0.57	0.38	0.87	0.55	0.38	0.79	0.40
Z Value	-	3.68 **	1.52	-	4.06 **	1.28	0.75	4.15 **	1.23	0.75	3.75 **	0.89
**Between-stages analysis:**												
**Social Skill**	**Cooperation**			**Empathy**			**Engagement**			**Communication**		
Comparison	A/B		B/C	A/B		B/C	A/B		B/C	A/B		B/C
Trend Change	/		\	/		\	/		\	/		\
	(+)		(-)	(+)		(-)	(+)		(-)	(+)		(-)
Change Between Levels	0–10%		95–65%	0–20%		85–85%	0–35%		100–100%	0–45%		88–85%
	(+10)		(−30)	(+20)		(=)	(+35)		(=)	(+45)		(−3)
Percentage Overlap	0.00%		100.00%	0.00%		100.00%	0.00%		100.00%	0.00%		100.00%
C Value	0.88		0.76	0.88		0.84	0.89		0.86	0.82		0.78
Z Value	4.65 **		4.07 **	4.69 **		4.55 **	4.72 **		4.63 **	4.34 **		4.21 **

** *p* < 0.01; S = Stable; US = Unstable. Note: A, B, and C denote the baseline, intervention, and maintenance phases, respectively. In the Trend row, “-” indicates no trend, “/” an increasing trend, and “\” a decreasing trend.

**Table 4 behavsci-15-01520-t004:** Visual analysis of Participant 2’s social skills.

**In-Phase Analysis:**												
**Social Skill**	**Cooperation**			**Empathy**			**Engagement**			**Communication**		
Sequence	A	B	C	A	B	C	A	B	C	A	B	C
Length	29	24	6	29	24	6	29	24	6	29	24	6
Trend	-	/	\	-	/	\	-	/	\	-	/	\
Trend Stability	100.00%	50.00%	50.00%	100.00%	54.17%	33.33%	100.00%	41.67%	33.33%	100.00%	62.50%	50.00%
Average	0.00%	62.50%	79.17%	0.00%	61.88%	75.00%	0.00%	82.08%	90.00%	0.00%	64.13%	77.17%
Level Range	0–0%	10–85%	70–85%	0–0%	20–85%	70–80%	0–0%	30–100%	85–95%	0–0%	10–88%	75–80%
Level Stability	100.00%	16.67%	100.00%	100.00%	41.67%	100.00%	100.00%	41.67%	100.00%	100.00%	12.50%	100.00%
	S	US	S	S	US	S	S	US	S	S	US	S
Level Change	0–0	10–85	85–70	0–0	20–85	80–70	0–0	30–100	95–90	0–0	10–88	80–75
	(=)	(+75)	(−15)	(=)	(+65)	(−10)	(=)	(+60)	(−5)	(=)	(+78)	(−5)
C Value	-	0.86	0.68	-	0.88	0.60	-	0.88	0.40	-	0.85	−0.01
Z Value	-	4.22 **	1.52	-	4.24 **	1.34	-	4.20 **	0.89	-	4.08 **	−0.03
**Between-stages analysis:**												
**Social Skill**	**Cooperation**			**Empathy**			**Engagement**			**Communication**		
Comparison	A/B		B/C	A/B		B/C	A/B		B/C	A/B		B/C
Trend Change	/		\	/		\	/		\	/		\
	(+)		(-)	(+)		(-)	(+)		(-)	(+)		(-)
Change Between Levels	0–10%		85–70%	0–20%		85–80%	0–30%		100–95%	0–10%		88–80%
	(+10)		(−15)	(+20)		(−5)	(+30)		(−5)	(+10)		(−8)
Percentage Overlap	0.00%		100.00%	0.00%		100.00%	0.00%		100.00%	0.00%		100.00%
C Value	0.93		0.86	0.91		0.88	0.90		0.87	0.87		0.85
Z Value	4.93 **		4.62 **	4.84 **		4.74 **	4.76 **		4.69 **	4.63 **		4.58 **

** *p* < 0.01; S = Stable; US = Unstable. Note: A, B, and C denote the baseline, intervention, and maintenance phases, respectively. In the Trend row, “-” indicates no trend, “/” an increasing trend, and “\” a decreasing trend.

**Table 5 behavsci-15-01520-t005:** Visual analysis of Participant 3’s social skills.

**In-Phase Analysis:**												
**Social Skill**	**Cooperation**			**Empathy**			**Engagement**			**Communication**		
Sequence	A	B	C	A	B	C	A	B	C	A	B	C
Length	53	24	6	53	24	6	53	24	6	53	24	6
Trend	-	/	\	-	/	\	-	/	\	-	/	\
Trend Stability	100.00%	37.50%	50.00%	100.00%	66.67%	50.00%	100.00%	50.00%	33.33%	100.00%	54.17%	50.00%
Average	0.00%	48.13%	72.50%	0.00%	36.46%	75.00%	0.00%	74.38%	87.50%	0.00%	54.42%	68.50%
Level Range	0–0%	0–85%	65–80%	0–0%	0–90%	65–85%	0–0%	30–100%	85–95%	0–0%	20–80%	60–75%
Level Stability	100.00%	25.00%	100.00%	100.00%	20.83%	83.33%	100.00%	20.83%	100.00%	100.00%	45.83%	100.00%
	S	US	S	S	US	US	S	US	S	S	US	S
Level Change	0–0	25–85	80–65	0–0	10–90	80–65	0–0	40–100	95–85	0–0	60–80	75–60
	(=)	(+60)	(−25)	(=)	(+80)	(−25)	(=)	(+60)	(−20)	(=)	(+20)	(−15)
C Value	-	0.60	0.68	-	0.58	0.58	-	0.73	0.60	-	0.58	0.54
Z Value	-	2.89 **	1.52	-	2.76 **	1.30	-	3.48 **	1.34	-	2.76 **	1.20
**Between-stages analysis:**												
**Social Skill**	**Cooperation**			**Empathy**			**Engagement**			**Communication**		
Comparison	A/B		B/C	A/B		B/C	A/B		B/C	A/B		B/C
Trend Change	/		\	/		\	/		\	/		\
	(+)		(-)	(+)		(-)	(+)		(-)	(+)		(-)
Change Between Levels	0–0%		85–65%	0–10%		90–85%	0–40%		100–95%	0–60%		80–75%
	(=)		(−20)	(+10)		(−15)	(+40)		(−5)	(+60)		(−5)
Percentage Overlap	4.17%		100.00%	16.67%		100.00%	0.00%		100.00%	0.00%		100.00%
C Value	0.69		0.60	0.65		0.58	0.78		0.73	0.66		0.57
Z Value	3.63 **		3.24 **	3.49 **		3.09 **	4.12 **		3.91 **	3.51 **		3.09 **

** *p* < 0.01; S = Stable; US = Unstable. Note: A, B, and C denote the baseline, intervention, and maintenance phases, respectively. In the Trend row, “-” indicates no trend, “/” an increasing trend, and “\” a decreasing trend.

## Data Availability

The original contributions presented in this study are included in the article. Further inquiries can be directed to the corresponding author.

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
