# Peer review of "The Impact of STEM Activities on Social Skills and Emotional–Behavioral Outcomes in Students with Autism Spectrum Disorder"

_behavsci, 2025, doi:10.3390/bs15111520_

Round 1
Reviewer 1 Report
Comments and Suggestions for Authors
Thank you for the opportunity to review this manuscript. While better understanding the impact of STEP activities on the social skills and emotional-behavioral outcomes of students with autism is important, this study contain flaws in the measurement system that call its findings into question.
Researchers indicated that they completed direct observation of the three participants for only one out or every ten minutes, and these data were reported in the multiple baseline graph. But the "target behaviors" are not actually operationalized for direct observation, and the examples that are provided do not provide sufficient clarity about what was observed. For example, the operational definition for "cooperation" indicates the student will "coordinate their own behaviors, and thereby achieve collective goals." The example suggests that the student follows a rule, but appears to infer that she would "accept the rule" (i.e., it is not possible to observe that a person "accepts" anything). Similarly, the example under "social withdrawal" suggests that particpants would need to move theri head to avoid eye contact for more than one minute, but if observers only record for one out of every ten minutes, then they would (by their own definition) miss most instances of this behavior.
Reviewer 2 Report
Comments and Suggestions for Authors
Thank you for the opportunity to review the manuscript titles, The Impact of STEM Activities on Social Skills and Emotional-Behavioral Outcomes in Students with Autism Spectrum Disorder. This paper addresses an important topic, as educators and researchers are increasingly seeking effective, evidence-based approaches to enhance both the social and emotional functioning of students with ASD within inclusive and specialized learning environments. The integration of STEM project-based learning with established intervention strategies is highly relevant given the current emphasis on 21st-century skills, interdisciplinary instruction, and innovative pathways to foster social inclusion for neurodiverse learners. Below I present my feedback for each major section of the manuscript with the hope that they will help researchers to strengthen their paper:
Introduction: This section is written well. It provides a strong rationale for why STEM based interventions are a promising avenue for addressing social and emotional-behavioral challenges in students with ASD. It integrated the theory, prior empirical work, and specific mechanisms by which STEM activities may support learning and development. Literature is current and international in scope. This section also states the research questions. To strengthen this section, the authors should to the following:
- Provide a clearer theoretical model that ties together social skills, emotional regulation, and STEM engagement.
- Clarify how this study uniquely extends prior robotics-based or problem-based learning interventions.
- Add research questions for maintenance, generalization and social validity which is typical for single-case research.
Method: This section is detailed and allow for replication in most part. Description of participants, settings, instructional framework, and data collection procedures are appropriate. The authors described the single-case research procedures in detail including baseline, intervention, and maintenance phases. Reliability and fidelity checks are reported, and social validity measures are used. For further improvement, the methods could be enhanced by:
- Elaborating on how peer partners were trained or supported during sessions.
- Expanding the description of the SSIS-RS adaptations for this context.
- Clarifying whether the same staff member collected and analyzed data, and how potential bias was minimized.
- Revising the experimental design section to reflect what appears to be used in this study. It looks like a multiple baseline across participants design is used but authors claimed that it was multiple probe.
- Describing how social validity interviews were analyzed.
Results: I like that the authors supplemented the visual analysis with effect size estimation. Separate presentation of social skills and emotional-behavioral outcomes allows readers to clearly see the impact. Social validity results provide important perspectives. To make the results section stronger, the authors should:
- Summarize findings across participants in a synthesis paragraph before moving into detailed participant-level descriptions.
- Reduce redundancy in reporting tables and figures. There are many areas where table and text overlap.
- Provide clearer interpretations of statistical results (e.g., what a Tau-U of 0.95 means in practical terms). Rakap (2015) has clear guidelines/becnhmarks for interpretation.
- Rakap, S. (2015). Effect sizes as result interpretation aids in single‐subject experimental research: description and application of four nonoverlap methods. British Journal of Special Education, 42(1), 11-33. https://doi.org/10.1111/1467-8578.12091
- Add confidence intervals for effect sizes for all calculations.
Discussion: Authors interpret their findings in light of cognitive, behavioral, and educational theories and offered plausible explanations for the observed improvements. They connected their results to broader inclusion and STEM education goals. The discussion of emotional-behavioral mechanisms linking predictability and environmental control to improved regulation was insightful. The discussion could benefit from the following revisions:
- More explicitly highlighting the novelty of combining STEM-PBL with evidence-based practices for ASD.
- Adding reflection on cultural context (China) and its influence on feasibility or generalizability.
- Offering clearer directions for scaling this intervention in mainstream classrooms.
- Expanding slightly on how findings could inform teacher training or policy initiatives.
Reviewer 3 Report
Comments and Suggestions for Authors
The topic of the research is very important and is followed by a very interesting manuscript.
The theoretical part (introduction) is well structured and written. Nevertheless, I would suggest adding, prior to the part about STEM, some reference to the transdisciplinary approach in general and specifically about transdisciplinary approach and inclusion. You may find some information in Flavian, H. (2024). Transdisciplinary teaching in inclusive schools: Promoting Transdisciplinary Education for Learners with Special Needs. Springer Nature.
Please check again if no other studies conducted regarrding the implementaion of STEM among learners with ASD.
Methods: Your study is based on three participants. You must refer to it as “case study” and accordingly refer to in through discussion and conclusion.
Your research procedure is well designed and presented.
How many “stakeholders” did you interview? Later on, how did you analyze the data you collected through interviews?
Discussion: The study is very interesting and important, but you should emphasize the fact you based it on three participants (and not wait for the limitation part).
Round 2
Reviewer 1 Report
Comments and Suggestions for Authors
I appreciate the authors attempt to strengthen the measurement system. It is surprising that the reported data did not change in any way with new, operationalized definitions. Apparently the authors used their newly operationalized definitions intuitively in the first data analysis, even though they did not describe those definitions sufficiently in the previous draft.
Reviewer 2 Report
Comments and Suggestions for Authors
I reviewed revised version of this paper against the feedback I provided during the first round of review. I am satisfied with the revisions the authors have made and believe the manuscript has been improved.
Reviewer 3 Report
Comments and Suggestions for Authors
I have read the revised manuscript and I think that the authors conducted a very professional work while revising it. Well done!